# Important Factors Influencing Willingness to Participate in Advance Care Planning among Outpatients: A Pilot Study in Central Taiwan

**DOI:** 10.3390/ijerph19095266

**Published:** 2022-04-26

**Authors:** Wan-Ting Tsai, Chun-Min Chen, Ming-Cheng Chung, Pei-Yu Tsai, Yen-Tzu Liu, Feng-Cheng Tang, Ying-Li Lin

**Affiliations:** 1Department of Family Medicine, Changhua Christian Hospital, Changhua 500, Taiwan; 181038@cch.org.tw (W.-T.T.); 161478@cch.org.tw (M.-C.C.); 76934@cch.org.tw (P.-Y.T.); 144084@cch.org.tw (Y.-T.L.); 2Big Data Center, Changhua Christian Hospital, Changhua 500, Taiwan; 180714@cch.org.tw; 3Institute of Medicine, Chung Shan Medical University, Taichung 402, Taiwan; 4Department of Holistic Wellness, Mingdao University, Changhua 500, Taiwan; 5Post Baccalaureate Medicine, National Chung Hsing University, Taichung 402, Taiwan; 6Department of Occupational Medicine, Changhua Christian Hospital, Changhua 500, Taiwan; 106159@cch.org.tw; 7Department of Leisure Services Management, Chaoyang University of Technology, Taichung 413, Taiwan; 8School of Medicine, Kaohsiung Medical University, Kaohsiung 807, Taiwan

**Keywords:** advance care planning, advance directives, patient autonomy, end-of-life care

## Abstract

Advance care planning (ACP) and advance directives (ADs) ensure patient autonomy in end-of life care. The number of ADs made and followed in Taiwan is still lacking. This study aimed to determine the factors that influence the willingness to participate in ACP among outpatients in Taiwan. In this study, we conducted a cross-sectional survey based on convenient sampling methods. The questionnaire included questions about participants’ basic sociodemographic information, knowledge of ACP, and awareness of ACP. A total of 198 adults who were outpatients of a family medicine clinic in an affiliated hospital in Taiwan were recruited. The associations between each variable were evaluated using the χ^2^ test. The adjusted odds ratios (ORs) and 95% confidence intervals (CIs) were calculated using the logistic regression method to examine the influence of each variable on willingness to participate in ACP. Being happy and being a healthcare professional were positively correlated with ACP participation. A lack of ACP knowledge (OR = 0.30 in model A and OR = 0.42 in model C), valuing “Reducing families’ end-of-life decision-making burden” (OR = 2.53 in model B and OR = 2.65 in model C), and a “Belief in a good death” (OR = 4.02 in model B and OR = 4.10 in Model C) were the main factors affecting subjects’ willingness to participate in ACP. Knowing which factors influence willingness to participate in ACP helps in the promotion of ACP. Continuously educating both the general public and healthcare professionals strengthens knowledge about the right to autonomy, about its associated laws, and about the ACP process, and thus, programs should be created to provide this education. Additionally, taking into account the differences between cultures can be helpful.

## 1. Introduction

With increasing acknowledgement of the importance of patient autonomy, advance care planning (ACP) and advance directives (ADs) were developed to ensure that a patient can retain as much autonomy as possible [1,2,3]. ACP is a process that includes structured interviews, meetings, and written advance directives. ADs are documents that record one’s preferences and decisions for end-of-life care. The ACP process guides people in creating ADs, empowers those people in implementing their ADs, and can guide families and professionals in planning the most appropriate care when patients lose their decision-making capacity [1,4]. Since the 1990s, the Patient Self-Determination Act in the United States has established the importance of ADs in a patient’s end-of-life care. In Taiwan, patient autonomy regarding end-of-life care is enforced by the Patient Right to Autonomy Act, which is the first law regarding patient autonomy in Asia and was made following global trends in end-of-life care. According to the Patient Right to Autonomy Act, people with a full capacity to make decisions can establish ADs regarding their end-of-life care. Compared with do not resuscitate (DNR) orders, which can be signed by a patient or their family, ADs made via the ACP process can only be signed by the patient. Therefore, adults who have the full capacity to make decisions are encouraged to participate in their ACP.

The number of ADs made and followed is still lacking. Only around 30 percent of people in the United States and Australia create ADs [5,6], and even less people from the non-cancer population [7] and in Asian countries such as China and Japan create ADs [8,9]. In Taiwan, people expressed a high willingness to make decisions about their end-of-life care (up to 70%); however, the number of ADs actually documented is still low (lower than 10%) [10]. This low rate of adoption is caused by many factors, and more opportunities are needed to discuss the issue of death. Previous surveys focused on populations with cancer or chronic life-limiting illnesses in which age, gender, higher income, lack of ACP knowledge, culture, and poor relationship with others have been shown to be influencing factors in the willingness to participate in ACP [1,11,12,13,14,15]. However, these surveys did not focus on a general outpatient population with the full capacity to make decisions. Thus, the purpose of this study is to determine the most important factors that influence willingness to participate in ACP among outpatients in Taiwan.

## 2. Methods

### 2.1. Study Design and Participants

We conducted a cross-sectional study using convenient sampling methods. A total of 198 outpatients from a family medicine clinic in an affiliated hospital in Taiwan were recruited from December 2019 to January 2020. The exclusion criteria included people who were deemed incompetent in making their own decisions, patients younger than 20 years old, people with dementia and a CDR (clinical dementia rating scale) score > 2, patients at the terminal stage of their disease, or people who were in an irreversible coma or in a permanent vegetative state. Ethical approval was obtained from the Institutional Review Board of the affiliated hospital (IRB No. 191014). The investigators developed a questionnaire in collaboration with the clinical researchers. Potential participants were approached by a trained doctor who explained the ACP process and the purpose of our study and distributed the questionnaire. All respondents signed informed consent forms before being given the questionnaire.

### 2.2. Instrument

This questionnaire was adapted from a pilot study that measured a participant’s attitude towards ACP [16]. Based on valid samples collected in that study, the reliability coefficient, Cronbach’s alpha, was >0.8. The questionnaire included questions about a participant’s basic sociodemographic information, knowledge of ACP, and awareness of ACP. Three categories of questions concerning their awareness of ACP were asked: the importance of participating in ACP, the willingness to participate in ACP, and reasons to participate or to not participate in ACP. The importance of participating and willingness to participate in ACP were measured by asking “How important do you think ACP is? (score 1 to 5: from very un-important to very important)” and “How willing are you to participate in ACP? (Score 0 to 10, with a higher score meaning higher willingness)”, respectively.

Some possible reasons for unwillingness to participate in ACP included the following: “Death is a taboo subject”, “Worried about being abandoned after signing advance directives”, “Do not understand ACP”, and “The ACP outpatient service charges a co-payment of TWD 3000” (yes or no) (TWD represents the New Taiwan Dollar; USD 1 is worth approximately TWD 30). Some possible reasons for willingness to participate in ACP included the following: valuing “Reducing families’ end-of-life decision-making burden”, a “Belief in a good death”, and a belief in ACP being “Legally guaranteed (The Patient Right to Autonomy Act guarantees the hope of hospice)” (yes or no).

Knowledge of ACP was measured by asking “Do you know that ’You have the right to refuse life-sustaining treatments’, ‘ACP should be carried out before writing advance directives’, ‘Advance directives need to be marked on your national health insurance card (NHI card) to be effective’, and ‘An ACP service can be co-paid’ (yes vs. no)”.

### 2.3. Outcome Variables

ACP willingness is a self-reported measure that focuses on a patient’s willingness to participate in ACP. The ACP-W consists of one item, with a potential score from 0 to 10, with a higher score meaning higher willingness. The average score and median for willingness to participate in ACP were 6.19 and 6.00, respectively (SD = 2.85). Accordingly, the cut-off score for ACP-W was reasonably expected to be ≥7, so we classified participants with scores of 0–6 and 7–10 as having low willingness and high willingness, respectively.

### 2.4. Other Covariates

Age, sex (male vs. female), marital status (married vs. single/no spouse), education level (≤junior high school vs. high school vs. ≥college), self-reported economic status (hard off vs. well off), self-reported mood (happy vs. unhappy), self-reported health (good vs. fair/poor), whether they were a healthcare professional (yes vs. no), and the reasons for participating in or not participating in ACP were the covariates used for the analyses.

### 2.5. Statistical Analysis

In this study, we analyzed the descriptive statistics of the subjects, their knowledge of ACP, and their willingness to participate in ACP. The associations between each variable were evaluated using the χ^2^ test. Using the logistic regression method, a multivariate analysis was conducted, and the adjusted odds ratios (ORs) and 95% confidence intervals (CIs) were calculated to examine the influence of each variable on the willingness to participate in ACP. IBM SPSS Statistics for Windows, version 22.0 (IBM Corporation, Armonk, NY, USA) was used to analyze the results. The level of significance was set at *p* < 0.05.

## 3. Results

Table 1 shows the demographic data and willingness to participate in ACP for 198 respondents. The distributions show that 55% of people were under the age of 50, more females (58%) than males (42%) responded to the questionnaire, 71% of the respondents were married, 75% had good self-reported economic status, and 71% had fair to poor self-reported health. Interestingly, a higher education level, self-reporting happiness, and being a healthcare professional were significantly associated with a higher willingness to participate in ACP.

### 3.1. Ranking the Importance of ACP and Reasons Affecting Willingness to Participate in ACP

Table 2 shows the evaluation of patients’ awareness of the importance of ACP and the reasons that may affect their willingness to participate in ACP. In total, most (138/198) respondents indicated that ACP is important to very important, of which 60% (85/138) showed a high willingness to participate in ACP. Compared with a low willingness, a high willingness to participate in ACP was significantly associated with the answers “Do not understand ACP”, “Reducing families’ end-of-life decision-making burden”, “Belief in a good death”, and “Legally guaranteed”.

### 3.2. Factors Influencing the Willingness to Participate in ACP

Table 3 shows the factors that affected ACP participation. Being happy and being a healthcare professional were positively correlated with participating in ACP. Respondents who were healthcare professionals were more willing to participate in ACP, with their adjusted ORs of ACP participation being 3.18 (95% CI = 1.48–6.83) in model A, 3.06 (95% CE = 1.44–6.51) in model B, and 2.81 (95% CI = 1.25–6.28) in model C. A lack of ACP knowledge was the main reason affecting subjects’ unwillingness to participate in ACP in both model A (OR = 0.30, 95% CI = 0.14–0.64) and model C (OR = 0.42, 95% CI = 0.19–0.92), while the reasons affecting respondents’ willingness to participate in ACP included “Reducing families’ end-of-life decision-making burden” (OR = 2.53 in model B and OR = 2.65 in model C) and “Belief in a good death” (OR = 4.02 in model B and OR = 4.10 in Model C).

### 3.3. Knowledge of ACP

Figure 1A,B displays the percentage of misconceptions about ACP among the general public and among healthcare professionals. For the general public, the most unclear parts were “An ACP service can be co-paid” (59%), followed by “ACP should be carried out before writing advance directives” (37%), “Advance directives needs to be marked on your national health insurance card (NHI card) to be effective” (32%), and “You have the right to refuse life-sustaining treatments” (17%). For healthcare professionals, the most unclear parts were “ACP should be carried out before writing advance directives” (30%), followed by “An ACP service can be co-paid” (27%), “Advance directives needs to be marked on your national health insurance card (NHI card) to be effective” (17%), and “You have the right to refuse life-sustaining treatment (LST)” (3%). In general, the general public knows considerably less about ACP than healthcare professionals. In addition, most of the public’s incorrect answers to ACP questions were attributed to “not understanding ACP”. According to these results, the two groups have inconsistent knowledge regarding ACP.

## 4. Discussion

In a survey of 198 outpatients from a medical center in Taiwan, 45% had a high willingness to participate in ACP, but actual participation was very low. People with non-medical backgrounds feel that participation in ACP is less important and are less willing to participate in ACP than healthcare professionals. Willingness to participate in ACP is influenced by a person’s awareness and knowledge of ACP. We further identify some barriers to ACP implementation, and the barriers are discussed one by one. Strengthening the understanding of ACP will be the first step to successfully implementing ACP in a community. Given that research on Taiwanese people participating in the ACP is sparse, our study can fill this research gap.

### 4.1. Factors Influence Willingness to Participate in ACP

#### 4.1.1. Different Professional Fields

About 70% of our participants think that ACP is very important, especially those who are healthcare professionals and have a relatively high willingness to participate in ACP. The main characteristics that increase agreement with and awareness of ACP are high education level, older age, and higher income [17,18,19]. In our study, 85% of healthcare professionals have a college degree or above, while 52% of the general public have an education level below high school. Different professional fields result in different opportunities to become educated about end-of-life care. Healthcare professionals have more opportunities to understand end-of-life decision-making, while the general public can only obtain such knowledge via relatives, friends, the news, and the Internet, meaning that having the same ACP and AD education programs for both groups is not suitable. We must adjust ACP and AD education for each group and use different methods to convey information to increase people’s understanding of ACP. For example, lectures and group discussions can be held for healthcare professionals, while posters, videos, TV commercials, lectures during religious activity, and even ACP online planning tools can be used for the general public [20].

#### 4.1.2. Happiness

Our survey identified that self-reporting happiness is significantly associated with a higher willingness to participate in ACP. Both openness and optimism are reported to be positively related to happiness [21,22], as patients who are more open are more willing to talk about their needs in future care [23]. A recent study showed that being optimistic and religious may decrease anxieties about death [21]. If patients are more open or optimistic, they may be willing to accept ACP and let their doctor know what kind of care they would like to have if they become unable to make medical decisions. However, healthcare professionals should also consider the impact of individual differences when promoting ACP. In addition, practitioners should develop their counseling skills to include optimistic aspects in order to reduce views that discussing end-of-life care is taboo. In short, people should be encouraged to remain optimistic about or open to discussing what kind of care they want if they have an illness they are unlikely to recover from.

### 4.2. Barriers to Advance Care Planning

#### Lack of ACP Knowledge and Awareness

A lack of knowledge about ACP is a barrier to engaging in or successfully implementing ACP that had been already mentioned by studies in the Western world and in Asia [7,12,13,24,25,26]. In our survey, for the general population, the main parts about ACP that are unclear are the ACP process and co-payment for ACP services. For healthcare professionals, the most unclear part is the ACP process. These results remind us that even if our government and hospitals promote ACP and ADs, nearly 40% of healthcare professionals still do not really understand the ACP process. Moreover, some healthcare professionals do not understand that they have “the right to refuse life-sustaining treatment”. We also found a relatively low proportion of people from the low-willingness group who think that ACP is important. Therefore, improving a person’s knowledge about both ACP and the right to autonomy will increase their willingness to participate in ACP. Thus, further educational programs to strengthen the knowledge about ACP are necessary to enhance their willingness to participate in ACP.

### 4.3. Facilitators in Implementing Advance Care Planning

#### 4.3.1. Role of Family-Centered Decision-Making Culture in ACP

From our survey, reducing families’ end-of-life decision-making burden is an important reason for people to participate in ACP. Confucianism has deeply influenced the Chinese culture, and thus, consideration of the family is a significant ethical consideration in medical decision making [27]. However, many families feel stressed when making end-of-life care decisions for their loved ones [28,29], and many families tend to choose life-sustaining treatments for patients to avoid being blamed for being unfilial [30,31]. If people can participate in ACP with their families and express their preferences for end-of-life care, their families will respect and defend their preferences, therefore potentially also reducing the burden of decision-making for the families. Thus, to reduce a family’s decision-making burden is a facilitator of willingness to participate in ACP [19,32].

#### 4.3.2. Belief in a Good Death

Belief in a good death also affects participation in ACP. A “good death” includes physical comfort, appropriate medical care, social support, and acceptance [33]. With the evolution of globalization and continued development of the Internet, patient autonomy has gradually become popularized within Asia. People have become aware that they may suffer if they are sustained by artificial nutrition/hydration or life-sustaining treatments. In addition, ACP can help people be cared for based on their preferred end-of-life care treatments, leading to a good death [11,34]. Discussing one’s end-of-life care decisions via ACP and recording one’s preference as ADs in an NHI card can guide medical professionals in planning the most appropriate care for a patient when they lose their decision-making capacity [1]. In our survey, “belief in a good death” is an important reason that significantly increases the willingness to participate in ACP. Thus, we can encourage patients to choose to be autonomous and to participate in ACP to avoid suffering during their end-of-life care and to achieve a good death. Furthermore, healthcare professionals should know the global trends regarding autonomy and should emphasize the importance of ACPs and ADs to prevent people from suffering during their end of life.

## 5. Limitation

Two limitations of this study should be noted. First, this study is descriptive by nature, using a face-to-face questionnaire survey method. Selection bias may be present due to the respondents being mainly from family medicine clinics in hospitals. In Taiwan, patients from family medicine clinics encompass the general population and are often healthy, which may represent the general population better than the other clinics or wards. However, generalization of the study results should still be met with caution. In addition, relying on closed questionnaires for objective data collection may compromise the accuracy of the data collected during face-to-face interviews by omitting qualitative data such as free comments. Finally, the factors that influence the willingness to participate in ACP and ADs completion are complicated, but we only discussed the influences of knowledge, family, and culture. Further studies that examine the influence of policies, religion, medical staff’s ability to guide a patient through ACP, and one’s experiences could be conducted in the future.

## 6. Conclusions

A lack of understanding of ACP remains a major barrier to both the general public and even healthcare professionals when creating ADs and implementing ACP. A high education level, a belief in and hope for a good death, and reducing the burdens put on families as they make end-of-life decisions are factors that facilitate the implementation of ACP, and integrating and promoting ACP will be easier if the participant’s culture is taken into account. ACP can be promoted in a more effective manner if further education is provided to the public and healthcare professionals to improve their understanding of the Patient Right to Autonomy Act and ACP procedures.

## Figures and Tables

**Figure 1 ijerph-19-05266-f001:**
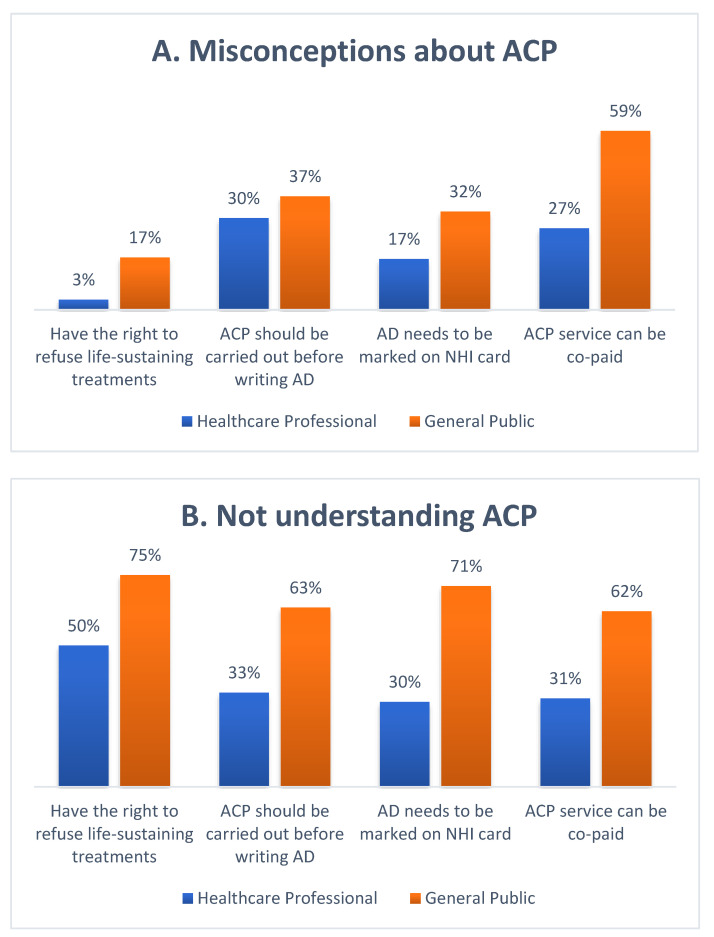
(**A**) Percentage of misconceptions of ACP between the general public and healthcare professionals. (**B**) Percentage of the general public and health professionals with misconceptions of ACP who think they do not understand ACP. (ACP: advance care planning; NHI: national health insurance; AD: advance directives).

**Table 1 ijerph-19-05266-t001:** General characteristics of respondents (*n* = 198).

□		Willingness to Participate in Advance Care Planning						□
		Low Willingness (0–6)		High Willingness (7–10)		Total		
□	□	*n* = 107		*n* = 91		*n* = 198		*p*-Value
Age	20–39	29	27%	31	34%	60	30%	0.594
	40–49	26	24%	24	26%	50	25%	
	50–59	28	26%	18	20%	46	23%	
	60–74	24	22%	18	20%	42	21%	
Sex	Male	49	46%	34	37%	83	42%	0.231
	Female	58	54%	57	63%	115	58%	
Marital status	Single/no spouse	29	27%	28	31%	57	29%	0.570
	Married	78	73%	63	69%	141	71%	
Education level	Junior high school or below	21	20%	5	5%	26	13%	<0.001
	High school	36	34%	18	20%	54	27%	
	College and above	50	47%	68	75%	118	60%	
Self-reported economic status	Hard off	27	25%	23	25%	50	25%	0.995
	Well off	80	75%	68	75%	148	75%	
Self-reported mood	Unhappy	70	65%	40	44%	110	56%	0.002 ^a^
	Happy	37	35%	51	56%	88	44%	
Self-reported health	Fair to Poor	81	76%	59	65%	140	71%	0.094
	Good	26	24%	32	35%	58	29%	
Whether they were a healthcare professional	Yes	18	17%	42	46%	60	30%	<0.001
	No	89	83%	49	54%	138	70%	□

^a^*p* < 0.01.

**Table 2 ijerph-19-05266-t002:** Awareness of ACP ^a^ and reasons affecting willingness to participate in ACP.

□	Willingness to Participate in ACP				□
	Low Willingness (0–6)		High Willingness (7–10)		
□	*n* = 107		*n* = 91		*p*-Value
Importance of ACP					
Very important	4	4%	36	34%	<0.001
Important	49	46%	49	46%	
General	35	33%	5	5%	
Unimportant	3	3%	1	1%	
Unable to answer	20	19%	0	0%	
Reasons affecting willingness to participate in ACP					
Death is a taboo subject	4	4%	4	4%	0.815
Worried about being abandoned after signing advance directives	17	16%	21	23%	0.200
The ACP outpatient service charges a co-payment of TWD 3000	49	46%	48	53%	0.329
Do not understand ACP	62	58%	24	26%	<0.001
Reducing families’ end-of-life decision-making burden	64	60%	79	87%	<0.001
Belief in a good death	58	54%	80	88%	<0.001
Legally guaranteed	27	25%	47	52%	<0.001

^a^ ACP: Advance care planning.

**Table 3 ijerph-19-05266-t003:** Factors influencing the willingness to participate in ACP ^a^.

□	□	Model A				Model B				Model C			
		Exp(B)	95% CI		*p*-Value		95% CI		*p*-Value	Exp(B)	95% CI		*p*-Value
			Lower	Upper		Exp(B)	Lower	Upper			Lower	Upper	
Age		1.00	0.98	1.03	0.894	1.00	0.97	1.03	0.902	1.01	0.98	1.04	0.565
Sex	Female	1.23	0.64	2.38	0.533	1.05	0.53	2.05	0.895	1.09	0.54	2.18	0.813
Marital status	Married	1.18	0.54	2.59	0.676	1.33	0.59	2.99	0.491	1.27	0.55	2.93	0.574
Self-reported economic status	Well off	0.63	0.28	1.40	0.252	0.67	0.29	1.54	0.341	0.66	0.28	1.55	0.340
Self-reported mood	Happy	2.74	1.27	5.89	0.13 ^b^	1.88	0.85	4.13	0.118	2.12	0.93	4.83	0.073
Self-reported health status	Good	0.99	0.45	2.17	0.976	0.99	0.45	2.15	0.970	0.90	0.40	2.05	0.807
Whether they were a healthcare professional	Yes	3.18	1.48	6.83	0.003 ^c^	3.06	1.44	6.51	0.004 ^c^	2.81	1.25	6.28	0.012 ^b^
Reasons to not	Death is a taboo subject	0.76	0.15	3.75	0.736					0.63	0.13	3.13	0.575
participate in ACP	Worried about being abandoned after signing advance directives	1.29	0.55	3.05	0.556					1.17	0.48	2.88	0.731
	The ACP outpatient service charges a co-payment of TWD 3000	0.68	0.33	1.41	0.299					0.50	0.23	1.10	0.084
	Do not understand ACP	0.30	0.14	0.64	0.004 ^c^					0.42	0.19	0.92	0.033 ^b^
Reasons to participatein ACP	Reducing families’ end-of-life decision-making burden					2.53	1.09	5.88	0.030 ^b^	2.65	1.07	6.52	0.034 ^b^
	Belief in a good death					4.02	1.75	9.22	0.001 ^c^	4.10	1.72	9.75	0.001 ^c^
□	Legally guaranteed	□	□	□	□	1.19	0.57	2.50	0.641	1.04	0.48	2.24	0.926

^a^ ACP: Advance care planning; ^b^
*p* < 0.05; ^c^
*p* < 0.01.

## Data Availability

The data that support the findings of this study are available from Changhua Christian Hospital. Restrictions apply to the availability of these data, which were used under license for this study. Dara are however available from the authors upon request and with permission of Changhua Christian Hospital.

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
