# Peer review of "Important Factors Influencing Willingness to Participate in Advance Care Planning among Outpatients: A Pilot Study in Central Taiwan"

_ijerph, 2022, doi:10.3390/ijerph19095266_

Round 1

Reviewer 1 Report

 This study aimed to investigate the factors that influence the willingness of ACP participation within outpatients. It is a cross-section survey study (N=198) using questionnaires for Advance care planning (ACP) and advance directives (ADs)   The association between variables was evaluated by χ2 test. The adjusted odds ratio (OR) and 95% 22 confidence interval (CI) were calculated by logistic regression method to examine the influence of variables on willingness to participate in ACP.

The paper is well written and the statistical analysis is also performed correctly. The presentation of the finding are clearly reported in the Tables and discussed in the text and in the limitation section.

The paper is worth publishing in this form, however I have a methodological issue to ask.

It would improve the paper if the authors include in the section which presents the instruments some account for validity and reliability, regarding the questionnaires and the specific measurements.

 This study aimed to investigate the factors that influence the willingness of ACP participation within outpatients. It is a cross-section survey study (N=198) using questionnaires for Advance care planning (ACP) and advance directives (ADs)   The association between variables was evaluated by χ2 test. The adjusted odds ratio (OR) and 95% 22 confidence interval (CI) were calculated by logistic regression method to examine the influence of variables on willingness to participate in ACP.

The paper is well written and the statistical analysis is also performed correctly. The presentation of the finding are clearly reported in the Tables and discussed in the text and in the limitation section.

The paper is worth publishing in this form, however I have a methodological issue to ask.

It would improve the paper if the authors include in the section which presents the instruments some account for validity and reliability, regarding the questionnaires and the specific measurements.

Author Response

Responses to Reviewer #1:

Review comment#1

The paper is well written and the statistical analysis is also performed correctly. The presentation of the finding are clearly reported in the Tables and discussed in the text and in the limitation section.

The paper is worth publishing in this form, however I have a methodological issue to ask.

It would improve the paper if the authors include in the section which presents the instruments some account for validity and reliability, regarding the questionnaires and the specific measurements.

Response to Reviewer comment No. 1:

We agree with the instrument information was not presented clearly enough. To better address this point, we made the revision in the section of instrument in present (revised) manuscript, as follows (Method section, page 2, para 4, line 84-86):

“2.2. Instrument

This questionnaire was adapted from a pilot study that measured a participant's attitude towards ACP [16]. Based on valid samples collected in that study, the reliability coefficient, Cronbach's alpha, was >0.8……

Reviewer 2 Report

Overall, a good paper with valuable information about the factors influencing the uptake of ACP in one country. These findings could be generalisable and some further literature may support this. My main concern with the paper is the English language - there are many instances that will need to be corrected. I recommend a professional editing service for this. 

The introduction provides good background information and helps to set the context.  The methods are generally well described, however, I do not see where the medical practitioners are included - was this a separate survye? Or were they part of the 198 participants? The results clearly show differences between medicos and non-medicos but I cannot find where they have been recruited as part of the study. 

The results are very well presetned - they align with the aims of the study and clearly define the findings in terms of the population characteristics. 

The discussion is fair - although some language editing would make it flow more easily. The sub-headings make it easy to see the main themes. The conclusions support the aims - well done. 

It is good to see the limitations described, although a further limitation would be the lack of qualitative data. Given that it was a face to face survey, it might have been useful to have a "free comment" part to gain further insight.  

Author Response

Responses to Reviewer #2:

Review comment#1

Overall, a good paper with valuable information about the factors influencing the uptake of ACP in one country. These findings could be generalisable and some further literature may support this. My main concern with the paper is the English language - there are many instances that will need to be corrected. I recommend a professional editing service for this. 

Response to Reviewer comment No. 1:

We apologize for the unprofessional expression in English, and the revised paper has undergone professional editing services. (Proofreading in red in the revised manuscript)

Review comment#2

The introduction provides good background information and helps to set the context.  The methods are generally well described, however, I do not see where the medical practitioners are included - was this a separate survey? Or were they part of the 198 participants? The results clearly show differences between medicos and non-medicos but I cannot find where they have been recruited as part of the study. 

Response to Reviewer comment No. 2:

Our apologies for the unclear information, and yes, they were among the 198 participants. in the survey, we asked outpatients about their occupations and further classified them into: with and without healthcare professional. The relevant information about this category is presented in Table 1. In order to avoid misunderstanding, we have changed "medical practitioner" in the text, tables and figures to "healthcare professional".

Review comment#3

The results are very well presented - they align with the aims of the study and clearly define the findings in terms of the population characteristics. 

The discussion is fair - although some language editing would make it flow more easily. The sub-headings make it easy to see the main themes. The conclusions support the aims - well done. 

It is good to see the limitations described, although a further limitation would be the lack of qualitative data. Given that it was a face to face survey, it might have been useful to have a "free comment" part to gain further insight.  

Response to Reviewer comment No. 3:

In responding to reviewer’s concern, we have modified the section of limitation, as follows: (Discussion section, page 8, para 3, line 273-276)

…”In addition, relying on closed questionnaires for objective data collection may compromise the accuracy of the data collected during face-to-face interviews by omitting qualitative data such as free comments.”……

Reviewer 3 Report

Dear All,

It was with great interest that I read the paper received for review.

While revising the publication, the following issues need to be clarified:

  1. Why did the authors of the study choose the participants over the age of 20 and not 18 years of age and older?
  2. The source materials have not been incorporated into the publication.
  3. The Discussion section does not include any references to the source materials.
  4. The Conclusion section needs to be shorter.
  5. A non-standardized tool was used while gathering the data for the paper.

Author Response

 Responses to Reviewer #3:

Review comment#1

Why did the authors of the study choose the participants over the age of 20 and not 18 years of age and older?

Response to Reviewer comment No. 1:

We appreciate the important point that the reviewer raises regarding the sample age considerations. The group of people over 20 years old was selected as the sample for this study because of the following reasons:

  1. In Article 12 of the current Civil Code, the age of majority has been revised from 20 to 18 as of January 30, 2022
  2. Regulations of the IRB require the consent of parents or legal guardians for research involving vulnerable groups (children under twenty years of age).

As the study period was 2019-2020, and the main purpose was to understand the attitudes of persons with capacity towards ACP, so the sample consisted mainly of adults over the age of 20.

Review comment#2

The source materials have not been incorporated into the publication.

Response to Reviewer comment No. 2:

We apologize for this omission, the source material has been placed in the revised version.

Review comment#3

The Discussion section does not include any references to the source materials.

Response to Reviewer comment No. 3:

We apologize for this omission, the source material has been placed in the revised version.

Review comment#4

The Conclusion section needs to be shorter.

Response to Reviewer comment No. 4:

To address the reviewer’s concerns, we have made the changes in the conclusion section (page8, para4, line 282-289):

Conclusion

“A lack of understanding of ACP remains a major barrier to both the general public and even healthcare professionals when creating ADs and implementing ACP. A high education level, a belief in and hope for a good death, and reducing the burdens put on families as they make end-of-life decisions are factors that facilitate the implementation of ACP, and integrating and promoting ACP will be easier if the participant’s culture is taken into account. ACP can be promoted in a more effective manner if further education is provided to the public and healthcare professionals to improve their understanding of the Patient Right to Autonomy Act and ACP procedures.”

Review comment#5

A non-standardized tool was used while gathering the data for the paper.

Response to Reviewer comment No. 5:

We apologize for the unclear information about the instrument and have added more description in the Method section. (page 2, para 4, line 84-86):

“2.2. Instrument

This questionnaire was adapted from a pilot study that measured a participant's attitude towards ACP [16]. Based on valid samples collected in that study, the reliability coefficient, Cronbach's alpha, was >0.8……

Round 2

Reviewer 3 Report

Accept in present form